# The Role of Graphene Monolayers in Enhancing the Yield of Bacteriorhodopsin Photostates for Optical Memory Applications

Roma Patel [1], Gregory Salamone [2] and Isaac Macwan [2,*]

1 Department of Biomedical Engineering, University of Bridgeport, Bridgeport, CT 06604, USA; rompatel@my.bridgeport.edu
2 Department of Electrical and Biomedical Engineering, Fairfield University, Fairfield, CT 06824, USA; gregory.salamone@colorado.edu
* Correspondence: imacwan@fairfield.edu; Tel.: +1-(203)-254-4000

**Abstract:** Bacteriorhodopsin (bR) is a photoactive protein that has gained increasing importance as a tool for optical memory storage due to its remarkable photochemical and thermal stability. The two stable photostates (bR and Q) obtained during the bR photocycle are appropriate to designate the binary bit 0 and 1, respectively. Such devices, however, have limited success due to a low quantum yield of the Q state. Many studies have used genetic and chemical modification as optimization strategies to increase the yield of the Q state. Nonetheless, this compromises the overall photochemical stability of bR. This paper introduces a unique way of stabilizing the conformations of bacteriorhodopsin and, thereby, the bR and Q photostates through adsorption onto graphene. All-atom molecular dynamics (MD) simulations with NAMD and CHARMM force fields have been used here to understand the interactive events at the interface of the retinal chromophore within bR and a single-layer graphene sheet. Based on the stable RMSD (~4.5 Å), secondary structure, interactive van der Waals energies (~3000 kcal/mol) and electrostatic energies (~2000 kcal/mol), it is found that the adsorption of bR onto graphene can stabilize its photochemical behavior. Furthermore, the optimal adsorption distance for bR is found to be ~4.25 Å from the surface of graphene, which is regulated by a number of interfacial water molecules and their hydrogen bonds. The conformations of the key amino acids around the retinal chromophore that are responsible for the proton transport are also found to be dependent on the adsorption of bR onto graphene. The quantity and lifetime of the salt bridges also indicate that more salt bridges were formed in the absence of graphene, whereas more were broken in the presence of it due to conformational changes. Finally, the analysis on the retinal dihedrals (C11 = C12-C13 = C14, C12-C13 = C14-C15, C13 = C14-C15 = NZ and C14-C15 = NZ-CE) show that bacteriorhodopsin in the presence of graphene exhibits increased stability and larger dihedral energy values.

**Keywords:** bacteriorhodopsin; graphene; photocycle; retinal; molecular dynamics

---

## 1. Introduction

Bacteriorhodopsin (bR) is a widely explored photoactive protein for a variety of biomolecular electronics applications, including optical memory storage [1,2]. bR is a membrane protein found in the salt marsh archaeon, Halobacterium salinarum, and comprises a photoactive moiety called retinal chromophore ($C_{20}H_{28}O$) in the middle of a channel formed by seven transmembrane alpha helices (A–G) [3]. It has a primary role of facilitating proton transportation and, therefore, generating a proton gradient across the cell membrane through a photochemical cycle [3]. This photocycle is a series of conformational changes caused by photoexcitation and de-excitation of covalently linked retinal chromophore atoms with the amino acid Lys216 via a protonated Schiff base linkage [1]. When bR is exposed to yellow light (~570 nm), it undergoes photoisomerization from an all-trans to

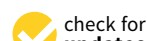



13-cis isomer and deprotonates by donating its extra hydrogen to the proton acceptor Asp85 amino acid [4,5]. This protonation causes a proton release to the extracellular surface from the Asp212, Arg82, Glu204 or Glu194 amino acids [4,6–12], after which a re-protonation occurs by the proton donor Asp96 amino acid [4,13]. This cyclic reaction leads to the formation of different conformational and photochemical intermediate states (photostates J, K, L, M, N and O) before returning to the ground state (bR) in around 4–5 ms. This results in a net translocation of one proton across the membrane (Figure 1a) [1,14].

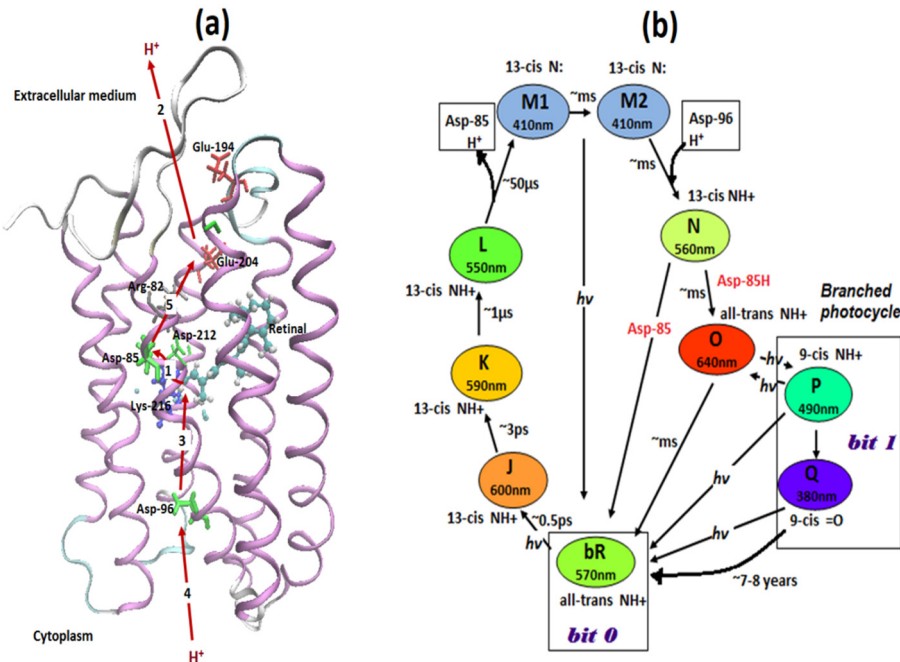

**Figure 1.** Proton transfer mechanism and binary logic in bR: (**a**) Location of key residues within bR and around the retinal showing the proton path from cytoplasm to extracellular environment; (**b**) Photochemical states of bR highlighting the 'Branched photocycle' that is responsible for binary state '1' (9-cis) for the molecule. Binary state '0' (all-trans) is also highlighted as the ground state.

The remarkable level of photochemical stability makes bR a powerful choice for optical memory storage devices. This stability is strongly correlated to its cyclicity, which is measured by the number of times the protein can undergo a photocycle before ~37% of it denatures (Figure 1b). Native bR protein has a cyclicity of ~$10^6$, which is higher than any other photochemical proteins (e.g., blue proteorhodopsin has a cyclicity of ~$10^4$) [14]. The second important property of bR in such an application is its thermal stability, as the protein has been observed to be structurally and functionally stable above 80 °C [15,16]. In order to store or record binary data, it is required to have two stable states. In bR, the ground (bR) state is appropriate to designate bit 0, as it remains stable until a yellow light activates the photocycle. Similarly, another stable state is the branched photocycle Q state, which can be assigned a bit 1 [1,14]. The protein enters the branched photocycle only when the all-trans O intermediate (where Asp85 is still protonated) is illuminated with red light (~640 nm). Here, bacteriorhodopsin photoconverts to an unstable 9-cis conformation called P state, which ultimately undergoes a thermal decay and forms free (hydrolyzed from Lys216) 9-cis retinal chromophore called the Q state [17]. This Q state is a highly stable intermediate state at ambient temperatures with a lifetime of 7–12 years and will only convert back to the bR state when it is exposed to near-UV light (~390 nm) [1]. These properties of the bR and Q states make the bR protein suitable for long-term memory storage devices, similar to semiconductor memories [1,14].

The challenge in realizing such a device comes in accessing the branched photostates to produce volumetric long term optical memory storage. The low yield of the O state (~3%) during the photocycle results in an even lower yield of the branched photoproducts (P

and Q) [1]. To overcome this limitation, many studies have focused on genetic engineering strategies to synthesize bR mutants with increased accessibility to the Q state [18,19]. Chemical optimization is another approach that is reported to achieve a higher yield of the O state and subsequent Q state. The addition of glycerol or organic cations (i.e., $Me_4N$ or $Et_4N$) and an adjustment of pH to 6–6.5 in a three-dimensional (3D) gel memory matrix inhibited proton release, which favored O state formation and ultimately increased O to P photoconversion [1,20]. Nevertheless, both techniques have significant disadvantages in which nearly all additional chemicals used affect the photochemical stability of the bR protein. Some genetic mutations caused structural changes to the active site, which altered other properties of the protein, for example bR mutants having lesser thermal stability compared to the wild type bR [1,2].

Alternatively, a new strategy is proposed here to increase and stabilize the formation of branched photoproducts by immobilizing bR protein on graphene nano-sheets. This approach is based on evidence that adsorption of proteins on nanomaterials through hydrophobic interactions induce conformational changes in the protein that can improve its functional properties [21,22]. Moreover, it also influences the intrinsic optical and electronic properties of the immobilized protein [21]. Molecular dynamics is utilized to understand such adsorption of bR onto the graphene sheet through non-binding forces and hydrophobic interactions. Quantification of the different properties of the bR–graphene complex, such as hydrogen bonds, root mean square deviation, salt bridges and the role of interfacial water molecules, was utilized to understand the efficacy of such a complex. Furthermore, secondary structure analysis of the bR protein in the presence and absence of graphene and dihedral energy versus dihedral angle analysis for the four major dihedrals of the retinal was performed in order to gather crucial insights into the stability of the bR–graphene complex and the respective photostates to make use of this system for volumetric optical memory devices. Finally, as we are considering the use of only monomeric bR to fabricate a memory device, the simulations we performed did not involve a lipid membrane. The goal of this study is to propose the use of graphene in place of a membrane and verify whether the adsorption of bR on graphene would in any way inadvertently affect the function of bR.

## 2. Materials and Methods

To study the behavior of the bacteriorhodopsin protein in the presence and absence of a monolayer graphene sheet, NAMD (Nanoscale Molecular Dynamics) software [23] integrated with a molecular visualization graphics software VMD (Visual Molecular Dynamics) was utilized [24]. In an all-atom isothermal and isobaric (NPT) ensemble of the NAMD software, 100 ns molecular trajectories were simulated for the bR control and experimental simulations (Figure S1). The all-atom system, containing the bR protein (230 residues) and the graphene sheet (3936 atoms), was structurally modeled using the CHARMM force field parameters available from the MacKerell lab at the University of Maryland [25]. The coordinate file (PDB) of the bR protein (1AT9) used in the study was acquired from the protein data bank [26]. The armchair graphene sheet with a size of 100 Å × 100 Å [27] was modeled using the graphene builder plugin in the VMD program. The structural parameters for the graphene sheet and those between the carbon C15 atom of the retinal chromophore and Lys216 were modeled using a topology manipulating plugin, TopoTools, and Molefacture [28,29]. The relative position of bR molecule was kept at ~15 Å from the surface of graphene to provide enough room between the molecules to ensure either adsorption or repulsion. Furthermore, the orientation was such that the retinal would be closest to the graphene sheet, even though it would not come in direct contact with graphene due to the presence of the surrounding amino acids. It was ensured that the adsorption occurs closest to the retinal site so that a direct effect of this adsorption can be captured on the dihedral angles of the retinal. Additionally, the state of the bR molecule was chosen such that it is in the ground state with the Schiff Base and ASP96 protonated. Both the systems (with and without graphene) were solvated using the TIP3 water model

containing 81,699 and 11,724 molecules, respectively [30], and a neutralizing concentration of sodium chloride was used to compensate for the overall net charge in the system. Additionally, based on the CHARMM topology parameters, there was no net charge modeled on bR or graphene molecules.

After setting the periodic boundary conditions, first, a 2500-step minimization using the steepest descent algorithm was carried out to remove the bad contacts and achieve minimum potential energy. This was followed by 500,000 steps of equilibration [31]. In all simulations, the temperature was maintained at 300 K by a Langevin thermostat and a pressure of 1 atm through a Nosé-Hoover Langevin piston barostat with a period of 100 ps and a decay rate of 50 ps. Multiple HP Z230 systems with eight cores, each using an Intel Xeon processor and a Quadro K620 CUDA acceleration capability, were used to perform all simulations. All simulations employed an integrated time step of 2 fs. A cut-off of 10–12 Å designated the short-range forces, while the long-range forces were calculated using the Particle Mesh Ewald (PME) algorithm. Root Mean Square Deviation (RMSD) and NAMD energy extensions were used to determine the interaction energy between bR and graphene. The VMD Timeline tool was used to analyze the secondary structure of the bR. TCL scripting was used to determine the number of atoms and interaction energies with respect to the cutoff distance from the surface of graphene and the optimal adsorption distance.

## 3. Results and Discussion

In order to understand the stability of the bR–graphene complex and to elucidate the structure–function relationship for the bR protein adsorbed on graphene, a detailed quantification of the interactive events at the bR–graphene interface is carried out and a role of water molecules is proposed. Typically, when a protein is adsorbed onto a hydrophobic surface such as graphene, it undergoes many conformational changes to adjust itself to the surface while maintaining its function. In fact, compared to those in the aqueous environment, proteins adsorbed on a surface have a greater potential to retain their stability and, hence, their function. This is primarily because of the fact that in an aqueous environment, a protein is constantly being targeted by the water molecules and ions from all sides of its largely hydrophilic surface. In the case where a protein is adsorbed on a surface, however, the protein exposes its core hydrophobic residues to strongly interact with the surface. This results in a less solvent-accessible surface area and, hence, a more stable protein. This binding between the protein and the hydrophobic surface is primarily due to the hydrophobic interactions and van der Waals forces, but it is envisaged that such interactions are also governed by the number of water molecules at the interface through the electrostatic energies. In this study, the stability of the protein and of the complex with graphene, as a whole, is quantified based on the root mean square deviation (RMSD), number and lifetimes of the hydrogen bonds at the interface, number and lifetimes of the salt bridges within the protein and the conformational changes of the protein via secondary structure analysis before, during and after the formation of the complex.

### 3.1. Stability Analysis of Bacteriorhodopsin in the Presence and Absence of Graphene

The RMSD data for the protein backbone over time were obtained to determine the stability of the protein and compared for bR in both the presence and absence of graphene (Figure 2a). Based on the RMSD curve, even though the deviations for bR in the presence and absence of graphene are similar, the fluctuations in the presence of graphene are much smaller as evident from the standard deviation (with graphene $4.1 \pm 0.2$ Å and without graphene $3.7 \pm 0.5$ Å), indicating greater stability. In the case of bR–graphene, RMSD increased by ~0.5 Å from 10 ns to ~65 ns, maintaining the stability as it adsorbed onto graphene. The peak at ~70 ns indicates a change in conformation of bR due to its increased interaction energy with graphene. On the other hand, RMSD, for the bR protein alone, shows continuous small fluctuations around a stable average value with less overall deviation. The RMSD of graphene (Figure 2b) maintains a consistent deviation of ~1.2 Å

with fluctuations of ~1.5 Å, indicating a fairly stable interaction with bR. Similarly, RMSD of the retinal showed a higher deviation in the presence of graphene compared to the control, indicating the effect the adsorption has on the diherals of the chromophore (Figure 2c). Finally, to analyze the behavior of bR in the vicinity of graphene, the distance between the centers of mass of bR and graphene was quantified. Over the entire period of the simulation, it was found that the centers of mass of bR and graphene came ~10 Å closer than in the beginning of the simulation, showing the attractive energies at play (Figure 2d).

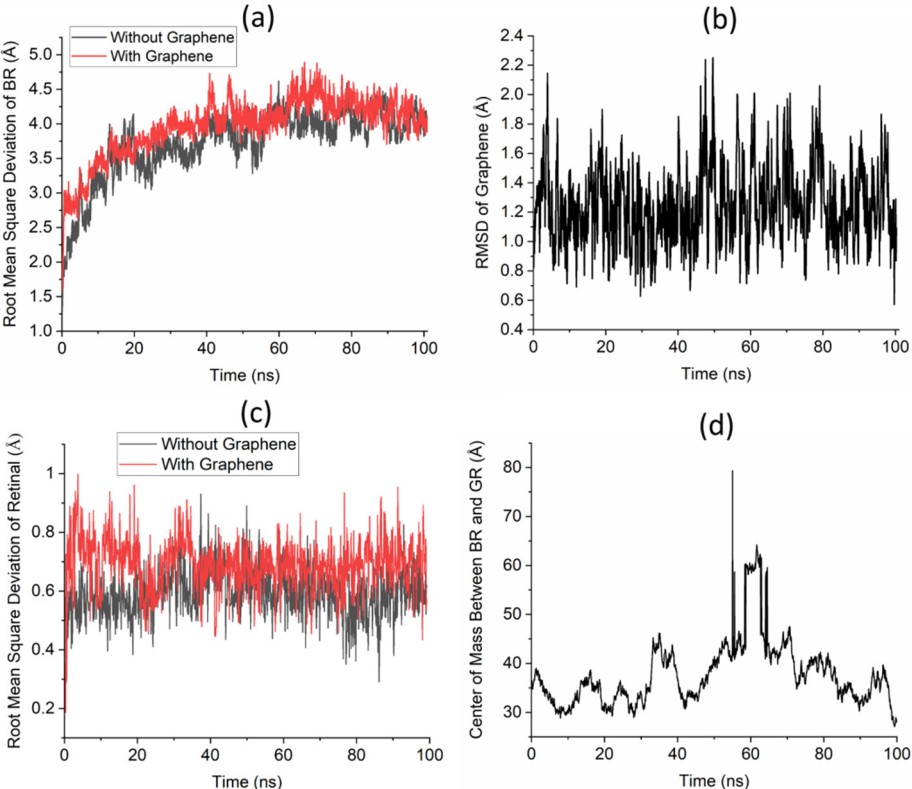

**Figure 2.** Stability analysis of bR and graphene during the 100 ns simulations: (**a**) Root Mean Square Deviation of bR with and without graphene; (**b**) Root Mean Square Deviation of graphene. (**c**) Root Mean Square Deviation of Retinal in the presence and absence of graphene; (**d**) Distance between the center of masses of bR and graphene.

### 3.2. Role of Hydrogen Bonds

It is known that hydrogen bonds help stabilize the protein molecule and, especially in bR, hydrogen bonds formed in the neighborhood of the retinal have a significant impact on the interface between bR and graphene. This paper is concerned with how this effect has a positive influence on the yield of the photocycle and its branched state. The criteria used for the hydrogen bonds involved a cut-off distance of 3.0 Å between the atoms and a cut-off angle of 20 degrees. The total number of H bonds formed by bR and within 5 Å of retinal with respect to time were analyzed. More H bonds were formed in bR–graphene (Figure 3b) compared to the bR control (Figure 3a) during the first 20 ns. However, for the rest of the simulation, the average number of H bonds formed were similar in both cases. The considerable variation of H bonds in bR–graphene during the initial period of interactions indicated the rapid conformational changes within bR exposing the non-polar amino acid residues to be adsorbed on the surface of graphene. To understand the spatial distribution of hydrogen bonds around the retinal site within the adsorption distance of 5 Å, the number of hydrogen bonds around the retinal active site were analyzed and found to be twice as much in the presence of graphene (Figure 3d) compared to bR control (Figure 3c) specifically after 40 ns. This reveals the time it took the adsorption onto graphene to affect the dihedral changes in the retinal. It is also known that water is a mediator in the effective

function of the bR protein, and as shown in Section 3.7, these hydrogen bonds around the retinal are also regulated by the number of water molecules in the vicinity of the retinal. A detailed 3D spatial distribution of the hydrogen bonds for the entire bR molecule in the presence and absence of graphene is presented in the supplementary Figure S7.

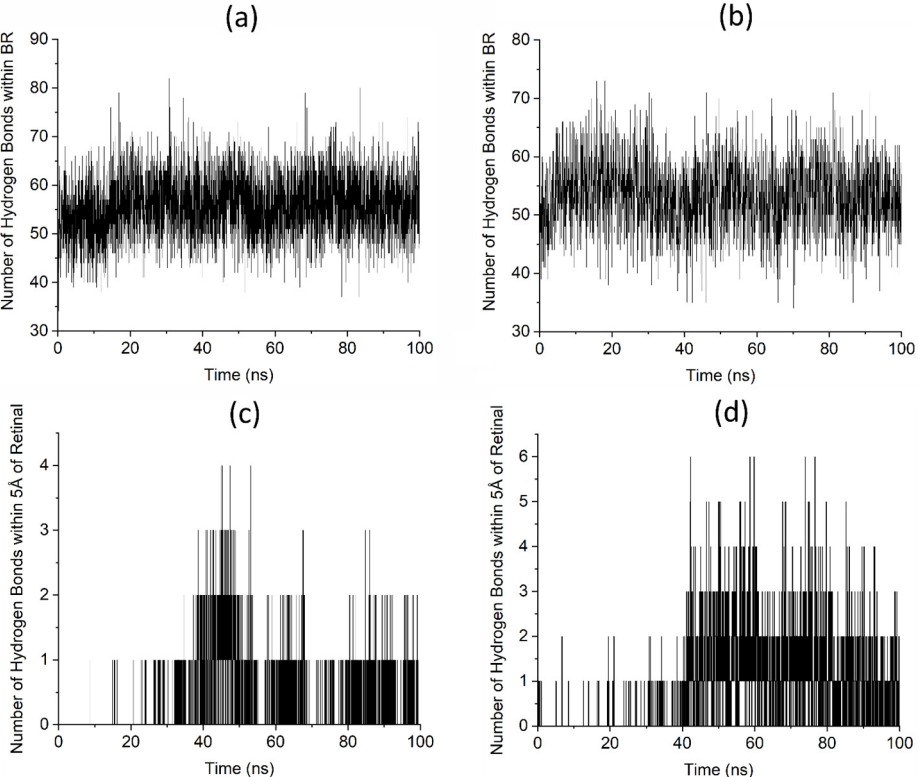

**Figure 3.** Role of hydrogen bonds in bR–graphene interactions: (**a**) Hydrogen bonds within bR in the absence of graphene; (**b**) Hydrogen bonds within bR in the presence of graphene; (**c**) Hydrogen bonds within 5 Å of the retinal in the absence of graphene; (**d**) Hydrogen bonds within 5 Å of the retinal in the presence of graphene.

### 3.3. Analysis of Salt Bridges in bR Protein to Determine the Stability of the bR–Graphene Complex

Salt bridges are an important form of electrostatic interactions that involve non-binding interactions between an oxygen atom of an acidic residue and the nitrogen atom of a basic residue. These not only determine the stability of but also contribute to the structure and function of a protein. Salt bridges were analyzed for both bR control (Table 1) and bR–graphene (Table 2), and the individual plots are as shown in Figure S2A,B. More salt bridges were formed in the case of bR control compared to the bR–graphene complex, whereas more salt bridges were broken for the bR–graphene complex compared to the bR control. This is because the protein undergoes large conformational changes exposing the internal hydrophobic residues to adsorb onto the graphene surface with greater energy and stability. Tables 1 and 2 also highlight the common salt bridges and their status during the control and experimental simulations. Although GLU166–ARG227 and ASP212–ARG82 salt bridges are sustained in both the control and experimental simulations, it is found that LYS216, which broke its interaction with ASP212 and formed one with ASP85 in the case of bR control, did the opposite in the presence of graphene. Here, LYS216 sustained its salt bridge with ASP212 and broke it with ASP85. It is important to highlight that the salt bridge ASP85–ARG82 in the case of the bR control is directly responsible for releasing the proton via ARG82 to trigger the conformational changes in the bR protein from photon absorption. As can be seen from Table 1, this salt bridge is formed and stays intact through the simulation (Figure S2A). In contrast to this, in the presence of graphene, it is found that the salt bridge between ASP85 and ARG82 breaks down (Table 2 and Figure S2B),

indicating that adsorption of bR onto graphene is capable of triggering conformational changes, similar to those of bR's photocycle. This, in turn, is hypothesized to create a favorable environment to increase the yield of the Q state and, hence, the binary bit '1' relevant to optical memory devices.

**Table 1.** Salt bridges within bR in the absence of graphene (the typical distance of 3.2 Å between Oxygen and Nitrogen atoms is considered as a criterion to determine the status of the salt bridges where distances less than 3.2 Å reflect formed salt bridges, greater than 3.2 Å reflect broken salt bridges and if the distances were lesser than 3.2 Å all throughout the simulation, it reflects the 'Sustained' category).

| Salt Bridges * | Broken | Formed | Sustained |
|---|---|---|---|
| **ASP96-LYS41** | ✓ | | |
| GLU194-ARG134 | | ✓ | |
| **ASP85-ARG82** | | ✓ | |
| **GLU166-ARG227** | | | ✓ |
| GLU9-ARG7 | | ✓ | |
| **ASP212-ARG82** | | | ✓ |
| **ASP21-LYS216** | ✓ | | |
| **ASP85-LYS216** | | ✓ | |
| **GLU194-LYS129** | | ✓ | |
| **ASP104-LYS159** | | ✓ | |
| **GLU166-ARG164** | | ✓ | |

* Data in bold represent the common salt bridges between control and experimental simulations.

**Table 2.** Salt bridges within bR in the presence of graphene (experimental simulations).

| Salt Bridges * | Broken | Formed | Sustained |
|---|---|---|---|
| **ASP96-LYS41** | ✓ | | |
| **GLU166-ARG227** | | | ✓ |
| GLU9–ARG7 | | ✓ | |
| **ASP212-ARG82** | | | ✓ |
| **ASP212-LYS216** | | | ✓ |
| ASP38-LYS41 | ✓ | | |
| **ASP104-LYS159** | | ✓ | |
| **GLU166-ARG164** | ✓ | | |
| **ASP85-LYS216** | ✓ | | |
| GLU74-ARG7 | | ✓ | |
| GLU204-ARG134 | ✓ | | |
| **ASP85-ARG82** | ✓ | | |
| **GLU194-LYS129** | ✓ | | |

* Data in bold represent the common salt bridges between control and experimental simulations.

ASP96, which is responsible for reprotonating the retinal during the final stages of the photocycle, can also be seen in Tables 1 and 2 (Figure S1A,B). In both the absence and presence of graphene, it is found that ASP96 loses its salt bridge with LYS41. Similarly, GLU194, which is also an important residue for proton transport, shows up forming a salt bridge with LYS129 in the case of bR control, whereas in the presence of graphene, it loses this bridge with LYS129. Finally, the residue GLU204 that is important for transporting

the proton to the extracellular medium through the proton diode gate at SER193 is seen to break its salt bridge with ARG134 in the presence of graphene (Table 2). The same ARG134, however, is seen forming a salt bridge with GLU194 in the absence of graphene (Table 1). This indicates that ARG134 may be the residue through which the proton is transported from GLU194 to GLU204 and then to the extracellular medium via SER193.

### 3.4. Secondary Structure of bR Protein

The secondary structure was analyzed using the timeline plugin for both the bR–graphene (Figure S3) and control bR (Figure S4). The timeline keeps track of secondary structure (turns, ß-sheet, α-helix, Pi-helix, isolated bridge, etc.) during all time steps and represents these different secondary structure types with different colors (Figure S5). Surprisingly, the bR secondary structure in the control group lost most of its ß turns as well as the two ß-sheets at the end of the simulation. bR appears to have slowly lost its secondary structure (Figure S4). Some changes were also observed in bR–graphene, including the loss of one or two turns, which was influenced by the graphene sheet during interaction and adsorption (Figure S3). However, unlike the control, it was observed that the overall stability of the secondary structure was maintained after adsorption. Moreover, the secondary structure at the active site amino acids has been also examined and compared. The α helices where active site amino acids Arg82, Asp85, Asp96, Asp212, Lys216 and Glu204 are present remained intact for the entire simulation in both control and bR–graphene. However, the ß turn where Glu194 is present near the cytoplasm was highly unstable for the control, while it remains stable for bR–graphene. Furthermore, in the secondary structure of a known 'O' state from the bR protein (3VIO) shown in Figure S6, it was also found that the residues 191 to 199 that make up the turns are highly unstable and at times conformed to coils in the absence of graphene. This indicates that the presence of graphene stabilizes the secondary structure at the active site in bR.

### 3.5. Energy of bR Protein

The total energy measured in the analysis comprises the non-bonding interaction energies (van der Waals and electrostatics) between graphene and the bR protein. The total energy value of bR protein in the presence of graphene decreases initially but then increases up to ~3150 kcal/mol at ~60 ns. After that, it remains stable around an average value of 2900 kcal/mol (Figure 4a). On the other hand, for bacteriorhodopsin without graphene, the total energy decreases gradually to 2600 kcal/mol. Comparing the two cases, the increase of ~500 kcal/mol in the presence of graphene is a clear indication of the adsorption of bR onto the surface of graphene. In regards to electrostatic energy, for bR–graphene it was ~400 kcal/mol lower than for the control, indicating that the adsorption of bR onto graphene is largely due to the VDW interactions (Figure 4b). In the case of VDW energy, there are two adsorption events that are observed at ~10 ns and ~70 ns, where the VDW energies are found to be ~90 kcal/mol and ~110 kcal/mol (Figure 4c). These energies together indicate changes in the conformational arrangements that are occurring in the bR protein. Such behavior in energy is also apparent in the key dihedral C14-C15 = NZ-CE, which is a double bond between the carbon C15 of retinal and the nitrogen NZ of the amine group of the LYS216 residue of bR (Figure 4d). In Figure 4d, the increased dihedral energy of bR–graphene suggests that it undergoes structural change due to the adsorption of bacteriorhodopsin onto graphene. Details of these dihedral analyses are given in the following Section 3.6.

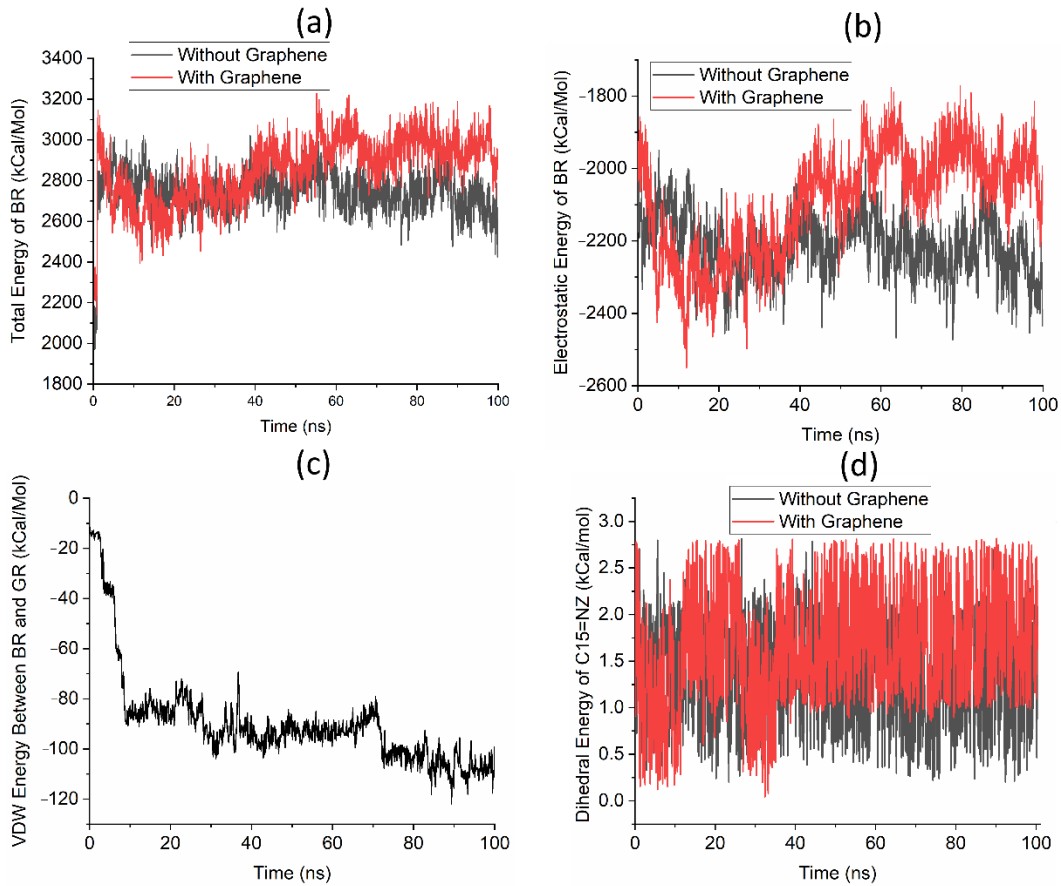

**Figure 4.** Analysis of different energies at the interface of bR and graphene: (**a**) Total energy within bR in the absence and presence of graphene; (**b**) Electrostatic energy of bR in the presence and absence of graphene; (**c**) Van der Waals energy at the interface of bR and graphene; (**d**) Dihedral energy of the key double bond between retinal and LYS216 of bR (C14-C15 = NZ-CE) in the presence and absence of graphene.

### 3.6. Retinal Dihedral Analysis

The retinal dihedral analysis involved investigating the trajectories of four retinal dihedrals (C11-C12-C13 = C14, C12-C13 = C14-C15, C13 = C14-C15 = NZ and C14-C15 = NZ-CE) for both bR–graphene and bR control. Dihedral data provide comprehensive insights into the structural changes of the retinal, and how it responds to the presence of graphene. By determining the energy values and angle values of the four dihedrals throughout the simulation, it was found that there is greater energy as well as stability in the dihedrals in the presence of graphene.

#### 3.6.1. Dihedral Energy Analysis

The trends in dihedral energy values were analyzed and it was evident that for the dihedral C12-C13 = C14-C15 (Figure 5b), there were greater energy values in the presence of graphene. This is most clearly visible in the case of C11 = C12-C13 = C14 dihedral (Figure 5d) and C14-C15 = NZ-CE dihedral (Figure 4d) introduced in Section 3.5. In the bacteriorhodopsin *Three State Model*, the protein reaches an energy barrier during photoexcitation to achieve a 90° twist around the C13 = C14 double bond and advance from ground state ($S_0$) to $S_1$ [32,33]. It is envisaged that with increasing rest state energies, observed in the bR–graphene dihedrals, it is easier to overcome this barrier as well as reach energy values that are required to undergo sufficient twisting for a complete photocycle.

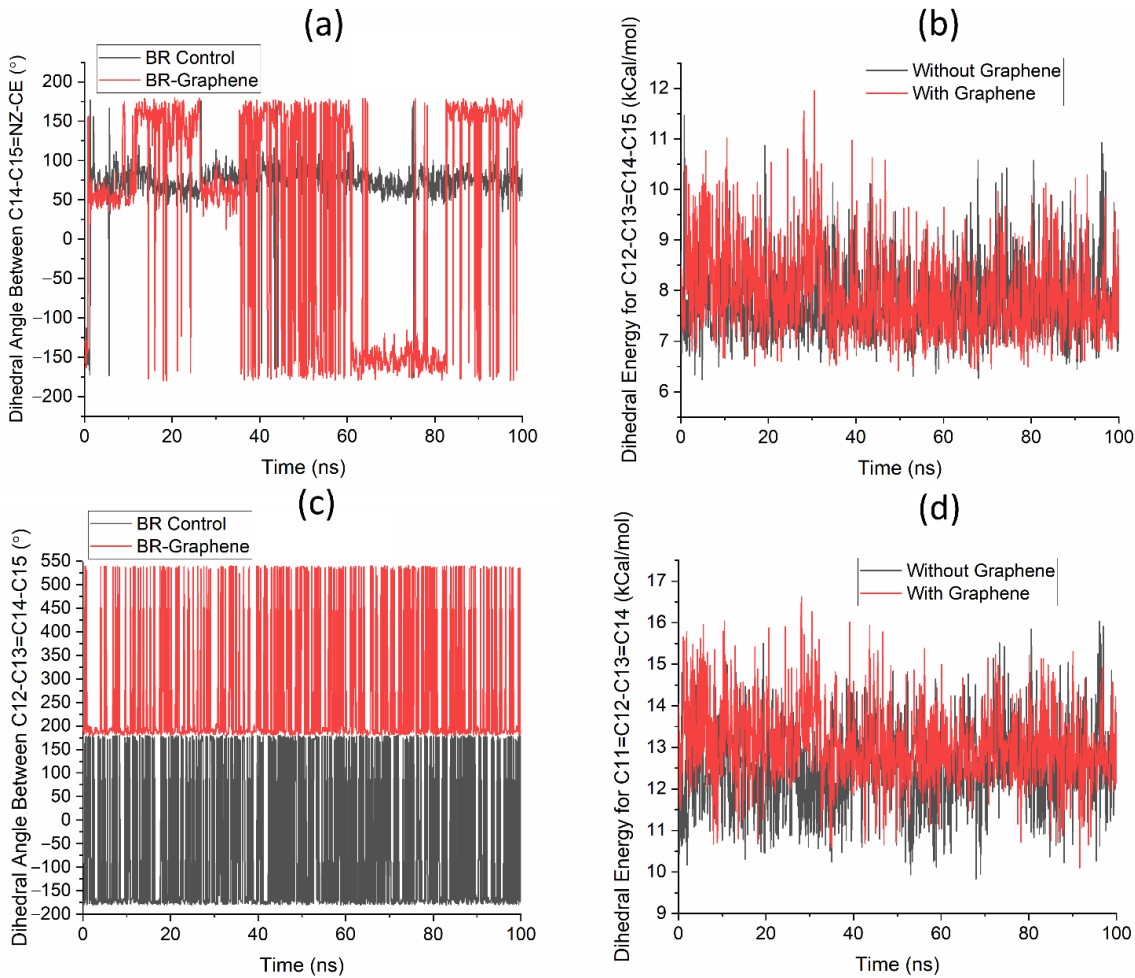

**Figure 5.** Analysis of Retinal Dihedral energies and angles: (**a**) Variation in the dihedral angle C14-C15 = NZ-CE between bR and retinal chromophore over a period of 100 ns; (**b**) Variation in dihedral energy for C12-C13 + C14-C15 dihedral within the retinal chromophore over a period of 100 ns; (**c**) Variation in the dihedral angle C12-C13 = C14-C15 within the retinal chromophore over a period of 100 ns (note the offset of 360 degrees for the bR–graphene plot to avoid overlap between the two plots in the presence and absence of graphene); (**d**) Dihedral energy changes for the dihedral C11 = C12-C13 = C14 over a period of 100 ns. All plots are compared for the presence and absence of graphene.

### 3.6.2. Dihedral Angle Analysis

The dihedral angle values are also responsible for portraying improved stabilization of bacteriorhodopsin. Figure 5a shows that the C14-C15 = NZ-CE dihedral angle of bR–graphene promptly reaches a range of ±120–180°, whereas the control appears to be stable around 60 degrees. These data indicate that 15 = NZ double bond for bR–graphene and control achieve trans (15-anti) and cis (15-syn) configurations, respectively [34]. Although it is curious that the control exhibits characteristics of a cis configuration in the ground state, it is important to highlight that, in contrast, graphene helps bR achieve a stable trans state, which is required in its ground state for a successful photocycle to occur. It is also worth mentioning that before reaching a *trans* state, the data in Figure 5a originally displays the bR–graphene 15 = NZ bond in a *cis* state (~60° angle), but it can be speculated that the adsorption onto graphene quickly corrects for this.

There is also evidence of stabilization in the bR–graphene C12-C13 = C14-C15 and C13 = C14-C15 = NZ dihedrals, however, C12-C13 = C14-C15, as seen in Figure 5c, is of stronger interest because of its role in the photocycle. More specifically, it is known that the rapid twisting of the 13 = 14 double bond is the mechanism of bR photoisomerization when the protein is exposed to light [35]. In Figure 5c, the bR–graphene is found to have a trend of rapid −180° to 180° dihedral angle switching, but to a lesser degree than the

bR control. Increased stability provided by graphene can ensure that the protein has proper conformation to achieve photoexcitation, and will immobilize the system as a whole. In summary, these dihedral angle observations along with the dihedral energy for key dihedrals provide an excellent comparison between the control bR and that adsorbed onto graphene in unexcited states. As discussed above, due to the adsorption of the protein onto graphene, not only is there a higher degree of stability, but also higher energy values that suggest a greater photocycle yield. A complete list of angles and dihedrals between the retinal and LYS216 is provided in the supplementary Table S1 and the parameters for the bonds, angles and dihedrals are provided in the supplementary Table S2.

### 3.7. Role of Water Molecules

For the first ~30 ns of the bR–graphene simulation, there were ~90 interfacial water molecules between bacteriorhodopsin and graphene. The number of water molecules then began to decrease at around 40 ns when bR was close to the graphene surface and ultimately stabilized at around 60 molecules once bR and graphene were at an optimal distance from one another (Figure 6a). The RMSD of the interfacial water molecules, however, was practically stable throughout the simulation (Figure 6b), which is a sign of stable adsorption of bR on the graphene surface and indicates that the interfacial water molecules were also crucial in regulating the interface. Comparing the presence and absence of graphene, there were more water molecules at the active site in the presence of graphene (Figure 6c) compared to the control (Figure 6d). From Figure 3d (Section 3.2), the number of hydrogen bonds around the active site also increased at ~40 ns in the presence of graphene, which can also be attributed to the favorable interactions between bR and graphene. Furthermore, it is understood that water molecules play a key role in the transportation of protons from one amino acid site to another during the photocycle. This is largely due to the hydrogen bonding of water molecules to the proton donor and acceptor groups [36]. As a result, more H bonds and increased water molecules around the active site means a greater probability that the photocycle is achieved.

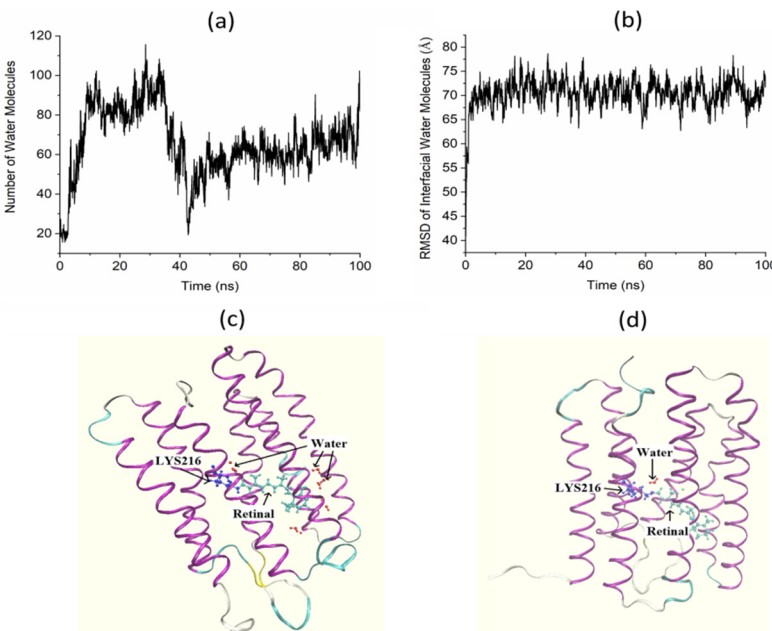

**Figure 6.** Role of water molecules at the interface of bR and graphene: (**a**) Number of interfacial water molecules between bR and graphene (within 5Å) during the 100 ns simulation; (**b**) Root Mean Square Deviation (RMSD) of interfacial water molecules between bR and graphene; (**c,d**) End of run screenshots showing the number of water molecules around the retinal in the presence and absence of graphene.

*3.8. Analysis of the Number of Atoms Adsorbed and the Optimal Distance of bR from the Surface of Graphene*

The distance between graphene and the retinal changes with time and reduces to ~22 Å from the initial distance of ~34 Å at 0 ns (a 12 Å approach due to the adsorption of bR on graphene surface) (Figure 7a). The conformational rearrangement of the retinal is the direct result of the adsorption of bR onto graphene. Moreover, an average number of bR atoms including hydrogen atoms (~11 atoms) are present at a 3 Å distance from the graphene surface. This number of atoms constantly increases as the cut-off distance increases up to 12 Å (Figure 7b). The attractive energy between these atoms and the surface of graphene also continuously increases with the distance ranging from 2 Å to 6 Å and becomes stable after 10 Å (Figure 7c). A comparison was done with and without hydrogen atoms to understand the interactions between heavy atoms (oxygen, carbon and nitrogen) as the heavy atoms are more stable. Figure 7d shows the optimal distance between the bR and graphene, where a minimum number of atoms interact with maximum energy and, hence, play a crucial role in stabilizing the interface. It was found that this optimal cut-off distance is ~4.2 Å.

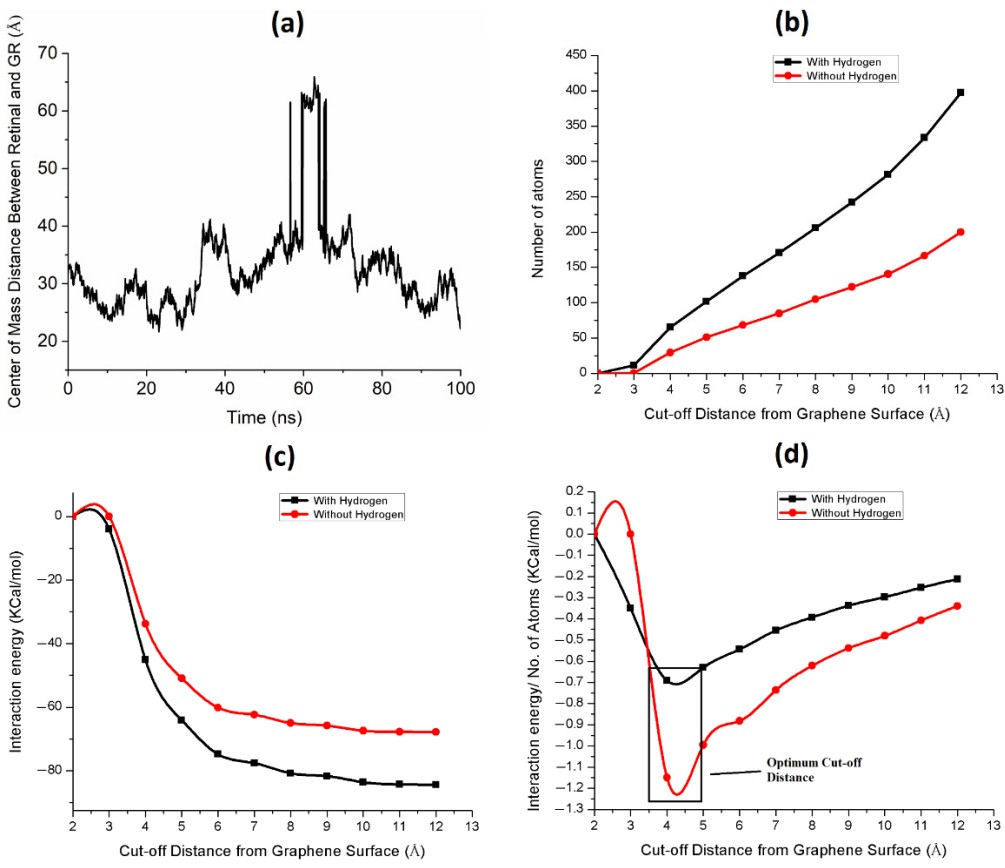

**Figure 7.** Optimal distance analysis between bR and graphene: (**a**) Distance between the center of the masses of graphene and retinal; (**b**) Number of atoms of bR with respect to cut-off distance; (**c**) Total interaction energy between bR and graphene with respect to cut-off distance; (**d**) Optimal distance between bR and graphene where minimum number of atoms drive the interactions at the interface. Note: in all plots, the data were gathered with and without hydrogen atoms to see the effect of heavy atoms (O, C and N) on the interface.

To further understand the nature of the interactions between graphene and bR, a thorough analysis was done to identify the type of residues adsorbed on the surface of graphene. As shown in Table 3, the interactions were largely due to the non-polar (hydrophobic) residues supporting the conformational changes in the bR and the resulting changes in the dihedral angles and energies.

**Table 3.** Amino acids (residues) of Bacteriorhodopsin adsorbed (within 5 Å) on the surface of Graphene.

| Residue Name | Residue ID | Residue Type |
| --- | --- | --- |
| Alanine | 2 | Non-polar, aliphatic |
| Isoleucine | 4 | Non-polar, aliphatic |
| Phenylalanine | 71 | Non-polar, aromatic |
| Glycine | 6, 72, 73 | Non-polar, aliphatic |
| Valine | 69, 130 | Non-polar, aliphatic |
| Tyrosine | 131, 133 | Non-polar, aromatic |
| Glutamine | 3, 75 | Polar, uncharged |
| Threonine | 5, 67 | Polar, uncharged |
| Proline | 70 | Polar, uncharged |
| Lysine | 129 | Polar, positive |

## 4. Conclusions

The adsorption of bacteriorhodopsin onto graphene has the potential to improve the yield of its Q photostate and, hence, its effectiveness for applications in optical memory devices. This notion is supported through both the stability and energetics data analysis. Higher deviation and less fluctuations in the RMSD data indicates more conformational changes that occur for bacteriorhodopsin due to its adsorption onto the graphene surface, and based on the graphene and bR interface study, it can be concluded that the optimal distance for these interactions to be maximized is ~4.2 Å. Due to the presence of graphene, a greater number of hydrogen bonds are formed around the retinal as a result of an increased number of water molecules and the regulation of the interface by these interfacial water molecules. This suggests the occurrence of conformational changes and the subsequent stability of the protein, which is also evident from the greater deformation of salt bridges in the presence of graphene, specifically in the deformation of the ASP85–ARG82 salt bridge. This results in the release of ARG82, which triggers the conformational changes in the bR and the retinal itself. Furthermore, the conformational changes are also found to be a byproduct of the increasing total and electrostatic energy of bR, which are deemed to be facilitators for the photocycle. Finally, the dihedral calculations for both angles and energies indicate greater stability and beneficial conformational changes of bacteriorhodopsin in the presence of graphene. In other words, rather than bR decaying to rest, graphene would help promote the advancement of photoisomer states and, in turn, it is postulated that this will mean a greater yield of the branched photocycle, the mechanism for binary data storage.

**Supplementary Materials:** The following are available online at https://www.mdpi.com/article/10.3390/app11209698/s1, Figure S1: Initial (0 ns) and Final (100 ns) orientations of bR and graphene, Figure S2: Salt bridges for bR control and bR—graphene simulations, Figure S3: Timeline of secondary structure of bR protein in presence of graphene, Figure S4: Timeline of secondary structure of bR protein in absence of graphene, Figure S5: Color key for secondary structure plot, Figure S6: Timeline of the secondary structure of the 'O' state of BR protein, Figure S7: 3D spatial distribution of hydrogen bonds within bR protein, Table S1: List of angles and dihedrals between retinal and LYS216, Table S2: Parameter for bonds, angles and dihedrals between retinal and LYS216.

**Author Contributions:** Conceptualization, I.M.; methodology, R.P. and I.M.; software, R.P. and I.M.; validation, I.M., R.P. and G.S.; formal analysis, R.P. and G.S.; investigation, R.P. and G.S.; resources, I.M.; data curation, I.M., G.S. and R.P.; writing—original draft preparation, R.P. and G.S.; writing—review and editing, G.S. and I.M.; supervision, I.M.; project administration, I.M. All authors have read and agreed to the published version of the manuscript.

**Funding:** This research received no external funding.

**Institutional Review Board Statement:** Not applicable.

**Informed Consent Statement:** Not applicable.

**Data Availability Statement:** Modeling and Simulation software used in this study, VMD (Visual Molecular Dynamics) and NAMD (Nanoscale Molecular Dynamics), is freely available from the Theoretical and Computational Biophysics group at the NIH Center for Macromolecular Modeling and Bioinformatics at the University of Illinois at Urbana-Champaign, http://www.ks.uiuc.edu/Research/vmd/ (accessed: 2 November 2018). Atomic coordinate (protein data bank, PDB) file for the bacteriorhodopsin protein, 1AT9, was obtained from the database located at the Research Collaboratory for Structural Bioinformatics (RCSB), www.rcsb.org (accessed: 31 August 2019). To create the protein structure files (PSF) and to carry out all-atom simulations, the necessary topology and force field parameter files were obtained from the Chemistry at Harvard Macromolecular Mechanics (CHARMM) database located at the MacKerell Lab at the University of Maryland, School of Pharmacy, http://mackerell.umaryland.edu/charmm_ff.shtml (accessed: 2 November 2019). The models for the two systems with and without graphene along with the parameter and configuration files are also available as part of the supporting information.

**Acknowledgments:** We thank Fairfield University and University of Bridgeport for providing computational resources to carry out and analyze the simulations.

**Conflicts of Interest:** The authors declare no conflict of interest.

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
