# Peer review of "The Role of Graphene Monolayers in Enhancing the Yield of Bacteriorhodopsin Photostates for Optical Memory Applications"

_applsci, doi:10.3390/app11209698_

Round 1

Reviewer 1 Report

              Bacteriorhodopsin (bR) is a light-driven H+ pumping membrane protein which exhibits photo-cyclic reaction. It is a highly stable molecule and can be photo-excited many times. Also, bR can be photo-converted to the near-UV absorbing (Q) state by second photo-excitation of the O-intermediate during the photocycle. This strong absorption change between the initial and Q states leads to the possibility of bR becoming a new memory storage medium material.

In this paper, Patel et al. studied the structural change and dynamics of bR induced by the adsorption onto graphene by all-atom MD simulation to gain new insights leading next-generation biomaterial based on bR to stabilize the structure of protein and to increase the efficiency of Q-state accumulation. Their results indicated considerable structural change of bR and optimal interaction distance between bR and a graphene sheet. However, there are many points where they lack a clear explanation of their simulation strategy; How did they set the relative position and orientation of bR against graphene at the beginning of the simulation? Don’t they need to consider different initial orientations? The results for the two systems (with and without graphene) are based on a single simulation each, but how was reproducibility assessed? Is the simulation time up to 100 ns long enough to equilibrate the system? bR usually takes a trimeric structure and interacts with lipids, but how well does their simulation of monomeric bR without lipid membrane the real system? To make these points clear, this reviewer recommends that the authors consider either a comprehensive and fundamental rewrite of the manuscript.

Other major and minor comments for further consideration by the authors are listed below.

-Major points

Materials and Methods

How was the protonation state of each residue treated? Since protonation state affects the hydrogen bonds and electrostatic interaction, the authors need to describe it in detail.

Page 6, 2nd paragraph, 8-9 lines

“However, for the rest of the simulation, the average number of H bonds formed were similar in both cases.”

Only the number of H bonds is difficult to understand the mechanism of the structural change of bR during the simulation. It is good to show the spatial distribution of increase and decrease of hydrogen bonds in the molecule.

Page 7, 1st paragraph, 5th-6th lines.

How were the break, formation and sustaining of salt bridges were categorized? Is there any quantitative criteria as the authors mentioned for H bonds?

-Minor points

Page 1, 1st paragraph, 1st line and 2nd paragraph, 9th line.

570-nm light is not green, but it is yellow. Also, ~390-nm light (2nd paragraph on the same page) is not blue, but it is near-UV light.

Page 2, 1st paragraph, 6th line

It would be better to include J and K to the photo-states as the author shown in Fig. 1b.

Page 3, 2nd paragraph, 3rd-6th line

“This approach is based on evidence that adsorption of proteins on nanomaterials through hydrophobic interactions induce conformational changes in the proteins leading to their structural and functional stability [21,22].”

In Ref. 21 and 22, structural change of protein upon adsorption on nanomaterials were reported, but this reviewer could not find the description about the protein stability.

Page 3, 3rd paragraph, 5th line

It would be better to full name of NPT here.

Page 5 1st paragraph 5-6 lines

Although the authors described that “the fluctuations in the presence of graphene are much smaller indicating improvement in the stability”, It is difficult to recognize the difference with only traces. More numerical evaluation (e.g. mean ± s.d.) would help to understand.

Page 7, Fig. 3 a-b

What is “H2 bonds” in the vertical axes?

Page 8, 1st paragraph, 5th-9th lines

“it is found that this salt bridge between ASP85 and ARG82 breaks down (Table 1b and figure S2B) indicating that adsorption of BR onto graphene is capable of triggering the conformational changes necessary to relevant changes in the retinal dihedral angle and energy. This in turn will also be a key factor in increasing the yield of the Q state and hence the binary bit ‘1’ relevant to optical memory devices.”

This part sounds like it is not strongly supported. The dihedral angle and the energy change do not necessarily increase the Q accumulation.

Page 8, 2nd paragraph, 1st-3rd lines

“ASP96 that initially receives the proton from the cytosol can also be seen in Tables 1a and 1b (figures S1A and S1B), however, in both the absence and presence of graphene it is found that it loses the salt bridge with LYS41.”

Many previous researches concluded that Asp96 is “protonated” in the dark state. Hence, the authors need to explain why they carried out the simulation with deprotonated Asp96 and how such simulation is informative to understand the real system.

Page 9, 2nd paragraph, 5th-6th lines

“Surprisingly, the BR secondary structure in the control group lost most of its ß turns as well as two ß- sheets at the end of the simulation.”

Why were the ß turns and two ß- sheets unstable in the simulation without graphene? These structures are considered to be very stable. This point might bring another concern about how the simulation without graphene correctly evaluated structural stability of the protein.

Page 11, Fig. 5a

Why did the 15=NZ bond of BR control take cis configuration? It is known that bR in the resting state takes trans configuration by the experimental researches.

Page 11, 2nd paragraph, 5th line.

Please captalize the “s” of “schiff” base the name is named after Hugo Schiff.

Page 12, 2nd paragraph 3rd-5th lines.

“The stability of C12-C13=C14-C15, as seen in Figure 5(c), however, is of stronger interest because it is the rapid twisting of the 13=14 double bond that is the mechanism of bR photoisomerization”

Why can this “double” bond be twisted even in the dark state? It is hard to understand without detailed explanation with more physicochemical point of view.

Author Response

We thank the reviewer for providing constructive comments and helping us improve the manuscript. We have addressed all the major and minor comments raised by the reviewer in the attached response.

Reviewer 2 Report

The author describe their work on the interaction of bR and graphene through absorption, and they found that the absorption can enhance the stability and greater dihedral energy, and then maybe increase the yield of the Q state of bR. The interaction of protien and graphene has been widely studied and many references have been reported. When bR was absorpted onto graphene, the conformation will be changed, and how the author can prove they still can keep their photochemical activities? From Figure 6 c and d, the comformation of bR has been changed significanlty. Also, How the author can prove the bR can produce more Q states after this absorption? I strongely recommend the author the compare the conformation of Q state bR and bR after absorption to show if the two conformation are similar.

Author Response

We thank the reviewer for their comments. We have addressed the points raised by the reviewer in the attached response.

Reviewer 3 Report

The paper is a complete theoretical/simulated study on the role of graphene in enhancing of bacteriorhodopsin photostates yield. It is an interesting work, well-structured and written.

I would suggest, in your possibilities, to make appealing the figures 3C and 3D, and figure 5. There are not easy for visualization.

Finally,do you think to look for experimental confirmations? 

Author Response

We thank the reviewer for the points raised. We have addressed the comments in the attached response.

Round 2

Reviewer 1 Report

              The authors gave point-by-point responses to the comment by this reviewer. However, unfortunately, their explanations for several points were not rational and convincing.

-“the specific placement of bR molecule is of little importance as it is all based on immobilization of bR on graphene during 100ns simulation.”

This reviewer cannot agree with their opinion. If the initial orientation of protein against graphene surface is different, it is possible that the interaction interface between bR and graphene and the resulting structural change in bR can be altered. It is mandatory to investigate that their finding is achieved with any initial molecular orientation.

-“In this study and others, we do observe that the simulation is reproducible”
It was not clearly shown in the paper. The reproductivity of protein dynamics is usually verified by repeating MD simulations with a different initial velocity of each atom, but no corresponding results were not shown in their manuscript and it is difficult why they can say that their simulation is reproducible.

-“As we are considering the use of only monomeric bR to fabricate a memory device, the simulations we performed did not involve a lipid membrane. The goal of this study is to propose the use of graphene in place of a membrane and verify whether the adsorption of BR on graphene would in any way advertently affect the function of bR.”

The usage of monomeric bR is quite different from the previous experimental researches as ref. 21 and 22 in which trimetric bR in lipid membrane was used. Also, the monomerization of bR highly reduces the protein stability and photo-reaction efficiency and affects the photocycle steps and so on. These points will make it difficult to attract strong interest from a broad range of readers. 

              Also, the preparation of bR without lipid or solubilizing detergent is experimentally highly difficult and not suitable for making a memory device, so that even from an application point of view, their simulation conditions look to be not so informative.

-The author replied that “Throughout the simulation, bR was in the ground state (without any net charge) and protonation occurs only when it gets energy from light so all hydrogen bonds and electrostatic interactions were analyzed in the ground state of bR in presence and absence of graphene”, but they seem to have failed to grasp the point of this reviewer's question.

Even in the ground state several residues (e.g. D96 and D115) are protonated, and the net charge of the entire protein molecule is not necessarily zero but it is compensated by ionic species in a solvent in the real case. Even if the authors intend to model the ground state of bR, the protonation state of each residue is mandatory to be considered carefully.

-“…we did perform the hydrogen bonds analysis surrounding the retinal molecule and reported the results in figures 3(c) and 3(d).”

Since the protein is an asymmetric molecule, it is not sufficient to understand the mechanism with showing the number of H bonds at a specific distance from the retinal. More detailed presentation of the 3-D distribution of hydrogen bonds as reported in https://pubs.acs.org/doi/abs/10.1021/acs.jpcb.0c02767.

-“We have rephrased the sentence to clear this misunderstanding “ASP96, which is generally observed as the proton receiver from the cytosol during proton transfer mechanism can also be seen in Tables 1a and 1b (figures S1A and S1B),…”

This point looks to contradict with the main purpose of their research in which they model the ground state of bR. D96 is deprotonated only in the photocycle and its deprotonated form cannot model the ground state in which D96 is always protonated (and net charge of the entire protein becomes zero in other parts of the molecule as mentioned above).

-“As supported in Ref 36, “Schiff Base Switch II Precedes the Retinal Thermal Isomerization in the Photocycle of Bacteriorhodopsin”, this indicates a cis configuration.”

Ref. 36 is about the N-intermediate, so that it does not support the current result of ground-state bR.

Author Response

We thank the reviewer for their comments and have uploaded our point by point response in the attached pdf. 
